# Roton-like acoustical dispersion relations in 3D metamaterials

Yi Chen 🆔 [1], Muamer Kadic[2,3] & Martin Wegener 🆔 [1,2]✉

Roton dispersion relations have been restricted to correlated quantum systems at low temperatures, such as liquid Helium-4, thin films of Helium-3, and Bose–Einstein condensates. This unusual kind of dispersion relation provides broadband acoustical backward waves, connected to energy flow vortices due to a "return flow", in the words of Feynman, and three different coexisting acoustical modes with the same polarization at one frequency. By building mechanisms into the unit cells of artificial materials, metamaterials allow for molding the flow of waves. So far, researchers have exploited mechanisms based on various types of local resonances, Bragg resonances, spatial and temporal symmetry breaking, topology, and nonlinearities. Here, we introduce beyond-nearest-neighbor interactions as a mechanism in elastic and airborne acoustical metamaterials. For a third-nearest-neighbor interaction that is sufficiently strong compared to the nearest-neighbor interaction, this mechanism allows us to engineer roton-like acoustical dispersion relations under ambient conditions.

[1] Institute of Applied Physics, Karlsruhe Institute of Technology (KIT), 76128 Karlsruhe, Germany. [2] Institute of Nanotechnology, Karlsruhe Institute of Technology (KIT), 76128 Karlsruhe, Germany. [3] Institut FEMTO-ST, UMR 6174, CNRS, Université de Bourgogne Franche-Comté, Besançon, France. ✉email: martin.wegener@kit.edu

For usual acoustical waves or phonons in gases, liquids, and solids, energy and momentum are proportional to each other[1]. Rotons can be seen as highly unusual acoustical waves with a parabolic minimum of the energy versus momentum at finite momentum and finite energy[2]. Based on a prediction by Landau[2] and following a suggestion by Feynman[3,4], the roton dispersion relation for longitudinal acoustical waves was observed in liquid Helium-4 (a Bose-liquid) at low temperatures by means of inelastic neutron scattering[5–7]. A bulk of later theoretical work interpreted rotons in terms of strong spatial correlations in this quantum system[8–10]. More recently, rotons have been found experimentally in two-dimensional (2D) liquid Helium-3[11] (a Fermi-liquid) and in Bose–Einstein condensates of erbium atoms subject to weak magnetic dipole–dipole interactions[12]. The roton dispersion relation has also been predicted theoretically for other ultracold dipolar[13] and quadrupolar[14] gases confined in one-dimensional (1D) and 2D geometries[15–17], as well as for Rydberg-dressed atoms[18].

The roton dispersion relation is illustrated in Fig. 1, where we represent energy as $\hbar\omega$, with the wave's angular frequency $\omega$, and (quasi-)momentum by $\hbar k$, with the wavenumber $k$. Apart from its basic physics, the dispersion relation in Fig. 1 is interesting for applications because it potentially allows to manipulate acoustical waves in unusual ways. First, it comprises a region of negative slope of $\omega(k)$ versus $k$, in which the wave's phase velocity $v_{ph} = \omega/k > 0$ and group velocity $v_{gr} = d\omega/dk < 0$ have opposite sign. Such behavior gives rise to backward waves and can lead to negative refraction at interfaces[19–21]. We will see below that this characteristic can occur without absorption/damping over a broad spectral regime. Second, the extrema of the dispersion relation in Fig. 1 correspond to zero group velocity and hence to peaks in the wave density of states. Third, at a given fixed angular frequency $\omega$ and for $k > 0$, the single dispersion curve supports three (one backward, two forward) wave modes with three different wavenumbers, phase velocities, and wavelengths. Unfortunately, no natural or rationally designed artificial materials showing roton-like dispersion behavior under ambient conditions have been reported so far.

It is well known that the Euler equation for classical gases or liquids[22] and the Navier equation for elastic solids[23] lead to dispersion relations of the type $\omega(k) = v_{ph}k$ for longitudinal or transverse acoustical waves in the bulk. Thus, neither of them captures roton-like behavior. In sharp contrast, it has recently been shown that chiral Eringen micropolar continuum elasticity theory[24] can lead to roton-like dispersion relations for transverse acoustical elastic waves[25]. In their work[25], chirality has been a necessary mechanism, whereas mechanisms based on periodicity, such as ordinary or extraordinary Bragg reflections, are not accounted for in micropolar elasticity theory[24]. More broadly speaking, mechanisms such as ordinary Bragg reflection[26,27], local resonances[28–31], near-ideal joints[32–34] introducing soft modes, spatial or temporal symmetry breaking[35–38], topology[39,40], duality[41,42], as well as geometrical nonlinearities[34,43] have independently given rise to a wealth of other unusual dispersion relations and quasi-static behaviors of elastic and acoustical metamaterials.

Here, we introduce and analyze a class of three-dimensional (3D) microstructured elastic metamaterials supporting roton-like transverse as well as longitudinal dispersion relations. We engineer these metamaterials by tailoring beyond-nearest-neighbor elastic interactions among the 3D metamaterial crystal unit cells in addition to the usual nearest-neighbor interactions. Furthermore, we apply the same concept to airborne acoustical waves in macroscopic three-dimensional channel-based metamaterials to illustrate the general nature of the approach.

## Results

**One-dimensional toy model.** Let us start our discussion by a simple 1D mathematical toy model illustrated in Fig. 2a: Identical masses $m$ separated by distance $a$ are connected to their immediate neighbors by linear Hooke's springs with spring constant $K_1$. In addition, each mass shall be coupled to the masses separated by distance $Na$ to the left and right (with integer $N \geq 2$) by springs with spring constant $K_N$. Newton's equation of motion for the acceleration $\ddot{u}_n$ of the mass displacement $u_n$ at

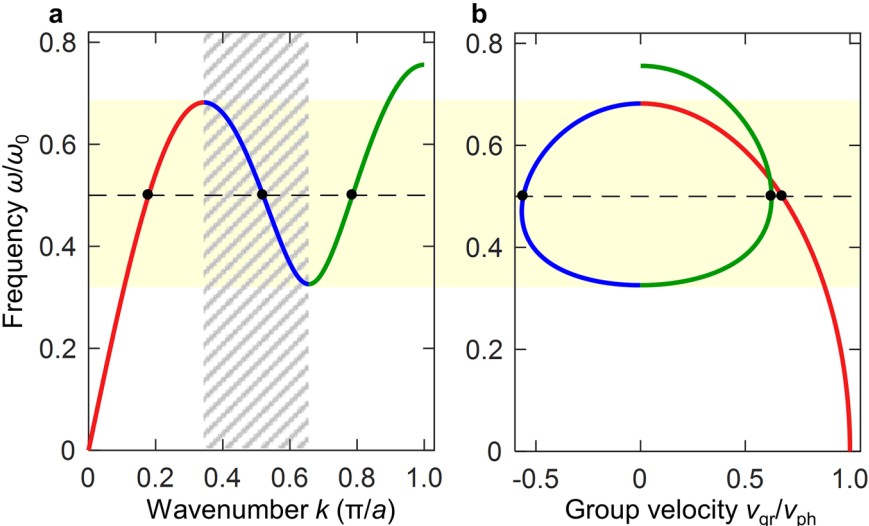

**Fig. 1 Roton-like acoustical wave dispersion relation. a** The acoustical wave's angular frequency $\omega(k)$ is depicted versus wavenumber $k$. The dispersion relation starts off with the usual linear increase of $\omega$ versus $k$. At a finite characteristic wavenumber, the dispersion relation exhibits a parabolic minimum. In a certain frequency range (highlighted by the light-yellow background), a single frequency $\omega$ leads to three modes at different $k$ (exemplified by the dashed line and the black dots), hence different wavelengths $\lambda = 2\pi/k$. In the hatched wavenumber interval, the group velocity $v_{gr} = d\omega/dk$ is negative, whereas the phase velocity $v_{ph} = \omega/k$ is positive. In a crystal with period $a$, the edges of the first Brillouin zone at wavenumbers $\pm\pi/a$ are important. **b** Corresponding group velocity versus angular frequency $\omega$, normalized by the phase velocity in the long-wavelength limit, $k \rightarrow 0$. The different colors serve to connect the different parts of the dispersion relation between **a** and **b**. Panels **a** and **b** can be taken as schemes. They actually correspond to solutions of the 1D toy model (cf. Fig. 2b) with parameters $N = 3$, $K_N/K_1 = 3$, and $\omega_0 = \sqrt{(K_1 + K_N N^2)/m}$.

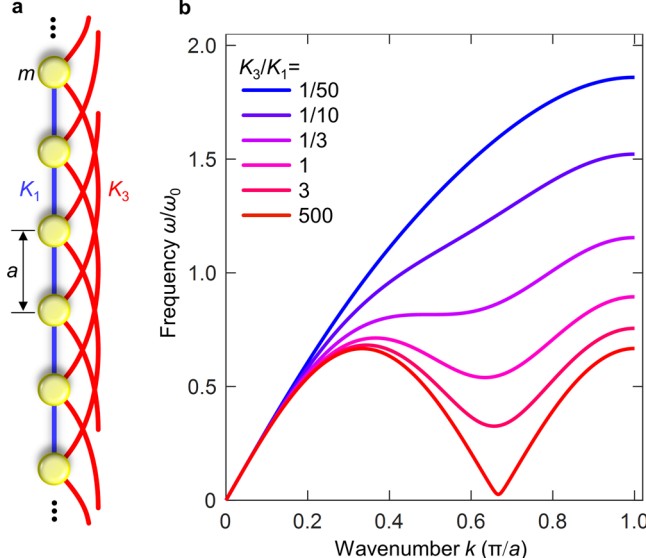

**Fig. 2 One-dimensional toy model. a** Masses $m$ (yellow dots) are connected to their nearest neighbors separated by distance $a$ by Hooke's springs with spring constants $K_1$ (blue straight lines). In addition, all masses are connected to their $N$th-nearest neighbors at distance $Na$ by springs with Hooke's spring constants $K_N$ (red curved lines). As an example, we choose $N = 3$. For $K_1 \neq 0$, the spatial period of this arrangement is $a$. Hence, the first Brillouin zone is given by wavenumber $|k| \leq \pi/a$. **b** Dispersion relation $\omega(k) = \omega(-k)$. The differently colored curves (see legend) represent different ratios of the spring constants $K_3/K_1$, increasing from top to bottom. For clarity, we fix the phase velocity in the long-wavelength limit $k \to 0$, i.e., $v_{\text{ph}} = a\sqrt{(K_1 + K_N N^2)/m} = \text{const} = a\omega_0$.

lattice site $n$ (with integer $n = -\infty, \ldots, \infty$) is given by

$$m\ddot{u}_n = K_1(u_{n+1} - 2u_n + u_{n-1}) + K_N(u_{n+N} - 2u_n + u_{n-N}). \quad (1)$$

The solution of this equation of motion are Bloch waves, $u_n(t) = \widetilde{u}\exp(\text{i}(kna - \omega t))$ with amplitude $\widetilde{u}$ and the imaginary unit i, following the dispersion relation

$$\omega(k) = \omega(-k) = 2\sqrt{\frac{K_1}{m}\sin^2\left(\frac{ka}{2}\right) + \frac{K_N}{m}\sin^2\left(\frac{Nka}{2}\right)}. \quad (2)$$

We emphasize that, for $K_1 \neq 0$ and $K_3 \neq 0$, the underlying spatial period is strictly $a$. Therefore, the borders of the first Brillouin zone lie at $|k| = \pm\pi/a$. For $0 \leq k \ll \pi/a$, we recover a usual acoustic-wave dispersion relation with $\omega(k) = v_{\text{ph}}k$ and phase velocity $v_{\text{ph}} = a\sqrt{(K_1 + K_N N^2)/m}$. For $N \geq 3$ and $K_N/K_1 > 1/N$, a minimum of $\omega(k)$ occurs inside of the first Brillouin zone at wavenumber $k \approx 2\pi/(Na) < \pi/a$.

Example dispersion relations for $N = 3$ and different ratios of $K_3/K_1$ (see different colors) are depicted in Fig. 2b. For $K_3 = 0$ and $K_1 \neq 0$, we obtain the usual acoustical phonon dispersion relation in the first Brillouin zone corresponding to spatial period $a$, i.e., $0 \leq k \leq \pi/a$. For the opposite limit of $K_1 = 0$ and $K_3 \neq 0$, the mass-and-spring model shown in Fig. 2a falls apart into $N = 3$ disconnected staggered one-dimensional chains, each with spatial period $Na = 3a$. We thus again obtain a usual acoustical phonon dispersion relation, however, the first Brillouin zone is given by $|k| \leq \pi/(3a)$. For $k \in [\pi/(3a), 2\pi/(3a)]$, the group velocity is negative, regardless of which Brillouin zone we use for the representation of the dispersion relation. The mean energy

flow (see Methods)

$$\langle p(k)\rangle = \frac{1}{2}\widetilde{u}^2\omega\left(K_1\sin(ka) + NK_N\sin(Nka)\right) \quad (3)$$

has two contributions. The contribution $\propto K_1$ is always positive for $0 < k < \pi/a$ and stems from the nearest-neighbor interactions. The contribution $\propto K_N$ due to the beyond-nearest-neighbor interactions is negative in the interval $k \in [\pi/(3a), 2\pi/(3a)]$, leading to a net negative energy flow if $K_N/K_1 > 1/N$. This aspect is analogous to what Feynman[4] referred to as the "return flow" in the context of the roton. Therefore, the sign of the group velocity and that of the energy flow are identical and independent on the choice of the Brillouin zone. The sign and magnitude of the phase velocity $\omega/k$, however, depend on the Brillouin zone. If and only if the nearest-neighbor interaction is finite, i.e., for $K_1 \neq 0$, the spatial period is $a$, hence $|k| \leq \pi/a$ is the proper first Brillouin zone. This leads to a positive phase velocity in the interval $k \in [0, \pi/a]$, whereas the group velocity and the mean energy flow are negative for part of this interval (approximately for $k \in [\pi/(3a), 2\pi/(3a)]$) for $K_N/K_1 > 1/N$. This behavior corresponds to a backward wave. The two wavenumbers for which the total energy flow and hence the group velocity are zero (cf. Fig. 2b) are merely special cases. On the basis of this discussion, the roton-like behavior can be seen as an unusual hybridization between two ordinary acoustical phonon dispersion relations, one for a mass-and-spring model with period $a$ and the other for a three-fold degenerate mass-and-spring model with period $Na = 3a$ (cf. Fig. 2b).

In other words, the roton-like minimum in the dispersion relation $\omega(k)$ of the 1D toy model with spatial period $a$ results from extraordinary Bragg reflections with reciprocal lattice vector $2\pi/l$, corresponding to the length $l = Na$ of the beyond-nearest-neighbor interaction. A roton-like minimum occurs if and only if the interaction has sufficiently long range (i.e., $N \geq 3$) and the strength of the beyond-nearest-neighbor interaction is sufficiently large (i.e., $K_N/K_1 > 1/N$). For $N \geq 4$, even several minima can occur within the first Brillouin zone.

To test our interpretation of the roton-like dispersion relation, we prescribe the displacement of a single mass in the middle of the 1D toy model chain (cf. Fig. 2a), $u_0(t) = \widetilde{u}\cos(\omega t)\exp(-(t/\tau)^2)$, by a temporal pulse with Gaussian envelope and carrier frequency $\omega = 0.5\omega_0$ in the spectral region of the roton-like dispersion relation for which $k(\omega)$ has three solutions. For the parameter range from $K_3/K_1 \approx 2$ to $K_3/K_1 \approx 5$, this excitation launches two clearly visible triplets of Gaussian wave packets (see Supplementary Figs. 1 and 2). This dependence on the ratio $K_3/K_1$ indicates that the effects of the discussed hybridization are most pronounced if the two ingredient phonon dispersion relations effectively have about equal weight. The right-going (left-going) triplet has positive (negative) mean energy flow and group velocity. Each triplet contains two forward waves and one backward wave. For the latter, group and phase velocity have opposite sign (see insets in Supplementary Fig. 1). These findings are consistent with the expectation from Fig. 1b and confirm our reasoning. Furthermore, Supplementary Fig. 3 shows that each of the three right-propagating modes can be excited selectively by tailoring of the excitation conditions.

**Three-dimensional microstructured elastic metamaterial.** Next, we translate the behavior of the 1D mathematical toy model into a practical metamaterial structure. From Fig. 2a it is clear that the (red) beyond-nearest-neighbor springs unavoidably overlap in two dimensions, making it necessary to go to three dimensions. The 3D architecture depicted in Fig. 3 (also see Supplementary

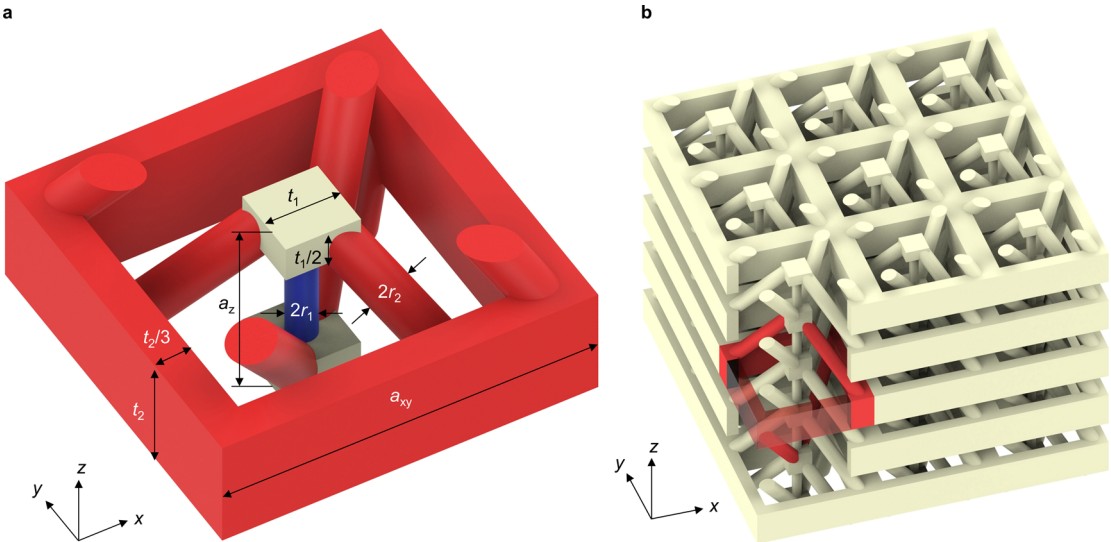

**Fig. 3 Designed three-dimensional elastic metamaterial structure. a** The architecture incorporates nearest-neighbor as well as beyond-nearest-neighbor interactions (cf. Fig. 2a) and is composed of a single ordinary linearly elastic constituent material. The colors are for illustration only. Elements mediating the elastic interaction between one layer and its third-nearest-neighbor along the $z$-direction are highlighted in red. The blue and red cylindrical rods have a radius of $r_1/a_z = 0.08$ and $r_2/a_z = 0.12$, respectively. The structure has no center of inversion but two mirror planes and a rotation-reflection symmetry, making it achiral and leading to a degeneracy of the lowest two transverse acoustical bands (cf. Fig. 4). The period of the structure along the $z$-direction is $a_z$, the corresponding first Brillouin zone edges lye at wavenumbers $k_z = \pm\pi/a_z$ (cf. Fig. 4). The period or lattice constant along the $x$- and $y$-directions is $a_{xy} = 2a_z$. The other geometrical parameters are $t_1/a_z = 0.40$, $t_2/a_z = 0.60$, and $a_z = 100\,\mu m$. **b** $3\times 3\times 5$ unit cells out of a corresponding bulk metamaterial, with the front corner cut out to allow for a view inside. The part highlighted in red illustrates the beyond-nearest-neighbor interaction. Two red rods connect a first cube to the red frame (made partly transparent at the corner). Two further red rods connect this frame to a second cube, which has a distance $3a_z$ with respect to the first cube. An animated view of the structure is given in Supplementary Movie 1.

Movie 1) is composed of a single ordinary linear elastic constituent material, e.g., a polymer. The masses in the 1D toy model are replaced by the small cubes with side length $t_1$. The effective spring constants of the nearest-neighbor (beyond-nearest-neighbor) interaction between these cubes are tailored by the radius of the thin (thick) cylindrical rods $r_1(r_2)$. The frame with height $t_2$ serves as an auxiliary structure to mediate the beyond-nearest-neighbor interaction for $N = 3$. Starting from any one cube, the oblique rods connect to the auxiliary frame, from which another set of oblique rods connects to the third-nearest-neighbor of the starting cube. Clearly, the two types of rods and the auxiliary frames introduce substantial additional mass, which needs to be considered. The resulting metamaterial structure in Fig. 3 is highly anisotropic, has no center of inversion, but two mirror planes and a rotation-reflection symmetry, namely a 90-degree rotation around the $z$-axis combined with a reflection of a plane parallel to the $xy$-plane. The structure is therefore not chiral and the lowest two transverse acoustical bands are degenerate by symmetry for propagation along the $z$-axis with wave vector $\vec{k} = (0, 0, k_z)$. When designing the elements mediating the beyond-nearest-neighbor interaction, it is important that the (local) resonance frequencies of these elements are pushed to much higher frequencies than the frequencies of the lowest-frequency acoustical bands. Otherwise, one obtains a complex band diagram comprising band crossings and avoided crossings such that the roton-like dispersion relation is obscured. Clearly, fulfilling this condition becomes increasingly difficult with increasing range of the beyond-nearest-neighbor interaction (i.e., with increasing integer $N$ in the toy model) because increasing length of beams clearly leads to decreasing beam resonance frequency at otherwise fixed parameters.

In Fig. 4a, we depict the calculated phonon band structure of the microstructure shown in Fig. 3. Here, we have numerically solved the eigenvalue problem for the Navier equation[23] for the displacement vector field $\vec{u}$ (see Methods). The used parameters for the ordinary elastic constituent material refer to a typical polymer with Young's modulus $E = 4.2$ GPa, Poisson's ratio $\nu = 0.4$, and mass density $\rho = 1140\,kg\,m^{-3}$. These parameters merely serve as an example. The frequency axis in Fig. 4 can easily be scaled to other values of $E$, $\rho$, and $a$ at fixed $\nu$. In Fig. 4a, we find roton-like dispersion relations for the two degenerate transverse acoustical bands as well as for the longitudinal acoustical phonon mode—as expected from our discussion of the 1D toy model. The higher bands plotted in gray are of lesser importance here. These bands emerging from $k_z = 0$ with finite $\omega$ do not occur in the toy model. They are partly due to local resonances of the long connecting cylindrical beams.

In Fig. 4b, we illustrate the energy flux along the $z$-direction in one unit cell for three different Bloch modes of the longitudinal band at the same frequency of 0.65 MHz. For the two eigenmodes **A** and **C** with positive group velocity, the mean energy flow in the vertical rods, acting as nearest-neighbor springs, and in the oblique rods, mediating the beyond-nearest-neighbor coupling, is positive. In contrast, for mode **B** with negative group velocity, the oblique rods support a backward propagating partial wave, while the partial wave in the vertical rods is a forward wave. The sum of the two energy flows is negative, consistent with negative group velocity. This behavior is the same as for the above 1D toy model. In both cases, the partial forward and partial backward energy flow lead to a vortex-like behavior of the energy flow. In his work on rotons[4], Feynman referred to the backward energy flow contribution as a "return flow".

The complete phonon band structure for all high-symmetry directions in three dimensions is shown in Fig. 5. Roton behavior only occurs along the $\Gamma$Z-direction (cf. Fig. 4a).

The roton-like behavior discussed thus far refers to the bulk, i.e., to a metamaterial crystal which is infinitely extended along all three spatial directions. It is interesting to ask whether the

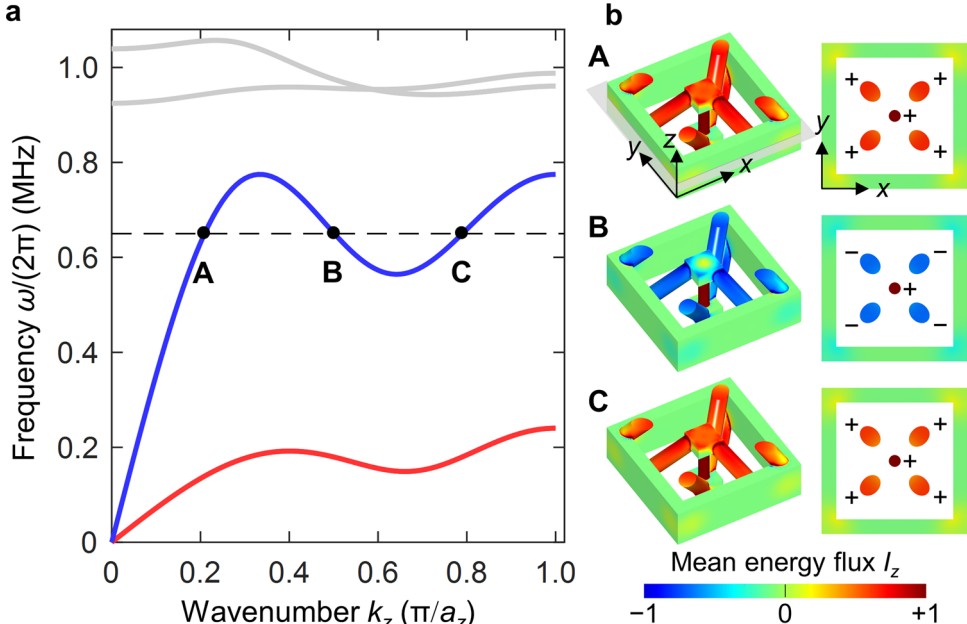

**Fig. 4 Elastic metamaterial phonon band structure along z-direction. a** The dispersion relation $\omega(k_z) = \omega(-k_z)$ of the architecture in Fig. 3 is shown for propagation of elastic waves along the z-direction with wavenumber $k_z$. The spatial period is $a_z$, corresponding to the first Brillouin zone given by the condition $|k_z| \leq \pi/a_z$ (cf. Fig. 1). The two transverse acoustical bands, which are degenerate by symmetry, are plotted in red, the single longitudinal acoustical band in blue. Higher dispersion branches are of lesser importance here and are depicted in gray. They partly result from local resonances within the unit cell, leading to finite values of $\omega$ at zero wavenumber $k_z = 0$. **b** Mean energy flux $I_z$ along the z-direction (on a false-color scale) corresponding to three eigenmodes marked as **A**, **B**, and **C** of the longitudinal band for the same frequency 0.65 MHz. The left column shows an oblique view of the unit cell, the right column a cut through the xy-plane. The mean energy flux in the thin vertical rods in the middle is positive for all three modes. The same holds true for the mean energy flow through the thicker oblique rods for modes **A** and **C**. The oblique rods mediate the beyond-nearest-neighbor interactions. In contrast, the energy flux through the oblique rods for mode **B** is negative, indicating a backward-wave behavior. Integration of the energy flux over the xy-plane for this mode also leads to a negative total energy flow, consistent with a negative group velocity. Parameters are $a_z = 100\ \mu m$ (cf. Fig. 3), aspect ratios as given in Fig. 3, Young's modulus $E = 4.2$ GPa, Poisson's ratio $\nu = 0.4$, and mass density $\rho = 1140\ kg\ m^{-3}$ for the constituent material.

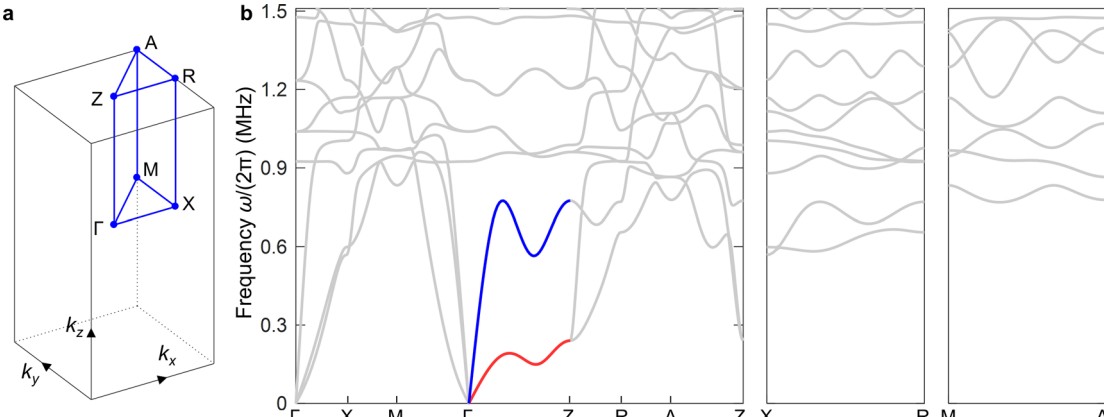

**Fig. 5 Elastic metamaterial phonon band structure in 3D.** As Fig. 4 (for the metamaterial structure shown in Fig. 3), but for many high-symmetry directions rather than only the $\Gamma Z$ or z-direction as in Fig. 4. **a** Illustration of the first Brillouin zone of the tetragonal-symmetry real-space lattice and selected high-symmetric directions in reciprocal space (marked in blue). **b** Calculated three-dimensional phonon band structure with the characteristic directions as indicated in **a**. Clearly, due to the used tetragonal symmetry, roton-like acoustical dispersion relations only occur for the $\Gamma Z$ direction. The corresponding colored bands (blue and red) are the same as the ones shown in Fig. 4.

behavior is robust and could also be observed in a metamaterial beam with finite cross section. Therefore, in Supplementary Fig. 4, we show results for a beam with a cross section of merely 2×2 unit cells (cf. Fig. 3a). A roton-like dispersion relation for the transverse bands is maintained. Roton-like behavior is also found for the twist band in Supplementary Fig. 4, which additionally appears due to the finite cross section of the beam.

Furthermore, we have emphasized that the structure shown in Fig. 3a, which contains the crucial beyond-nearest-neighbor interactions, is not chiral. Therefore, chirality is clearly not a necessary condition for roton-like behavior. Due to the absence of chirality, the two lowest transverse bands in Fig. 4a are degenerate and the associated eigenmodes do not contain any sort of rotation. However, we can introduce chirality into this metamaterial structure by "twisting the rods" mediating the

beyond-nearest-neighbor interactions. We additionally double the number of these rods to obtain four-fold rotational symmetry around the $z$-axis. The resulting structure is illustrated in Supplementary Fig. 5, which can be compared to its achiral counterpart in Fig. 3. The corresponding phonon band structure depicted in Supplementary Fig. 6 again shows a roton-like dispersion relation. In addition, as a result of chirality, the degeneracy of the two lowest transverse bands is lifted (compare Fig. 4a and Supplementary Fig. 6) and the eigenmodes become chiral (see Supplementary Fig. 6), which means that they are directly associated to rotations within the unit cells. This aspect makes the connection to the original interpretation of rotons according to Landau[2] and Feynman[3] in terms of rotations of groups of Helium-4 atoms even closer.

**Three-dimensional tube-based metamaterial for airborne sound.** We now translate the findings of the 1D toy model to airborne acoustical waves (or sound) rather than elastic waves in the previous section. Recall that classical forces (in the sense of Newton's second law), which are mediated by the Hooke's springs in Fig. 2a or by the cylindrical solid elastic beams in Fig. 3, can be interpreted as momentum currents[44,45] in the language of continuum mechanics. For airborne acoustical waves, the momentum current is directly related to the instantaneous air current. The air current along the tube axis in a cylindrical tube with rigid walls can obviously be controlled by the inner cross section of the tube.

This analogy allows us to propose a 3D metamaterial architecture for airborne sound. The structure is the complement of the one shown in Fig. 4a. This means that the constituent material is replaced by voids and vice versa. Air propagates in the resulting channels inside a rigid material. By tailoring the inner diameter of the channels, we engineer the effective strength of the nearest-neighbor and beyond-nearest-neighbor interaction, respectively. The calculated acoustical wave dispersion relation shown in Fig. 6a again exhibits roton-like behavior. Here we have

neglected friction between air and the walls (see Methods). The latter assumption has been used many times in the literature[46,47] and is justified if the diameter of the channels is sufficiently large. Therefore, we consider rather macroscopic parameters in Fig. 6 ($a_z = 10$ cm). As air exclusively supports longitudinal pressure waves (and no transverse modes), the resulting overall band structure in Fig. 6a is simpler than the one for elastic waves in Fig. 4. Finally, we depict examples of the energy flow in Fig. 6b. As expected from the mentioned analogy between forces and air currents, the behavior is closely similar to that shown in Fig. 4b for the 3D elastic metamaterial.

## Discussion

The famous roton dispersion relation for acoustical waves has first been predicted for and later observed in the Bose-liquid Helium-4 at low temperatures[2–6]. More recently, it has also been discussed for the Fermi-liquid Helium-3 at low temperatures[11] and for interacting atoms in Bose–Einstein condensates[12–17]. Here, we have realized roton-like dispersion relations by designed periodic metamaterials for both, elastic waves or phonons in solids and pressure waves in gases. All of these are classical systems operating under ambient conditions. The underlying mechanism is based on designed third-nearest-neighbor interactions in addition to the usual nearest-neighbor interactions. The third-nearest-neighbor interaction gives rise to a hybridization of phonon branches with different spatial periods and hence to extraordinary Bragg reflections with reciprocal lattice vectors smaller than the wave vector at the edge of the first Brillouin zone. For both, the proposed (achiral and chiral) 3D microscopic microstructures for elastic waves and the proposed 3D macroscopic channel-based structures for airborne sound waves, the 3D additive manufacturing technology required to make the metamaterial unit cells is readily available. However, large numbers of unit cells are needed to avoid edge effects. This aspect together with directly measuring the roton-like dispersion relations represents a challenge.

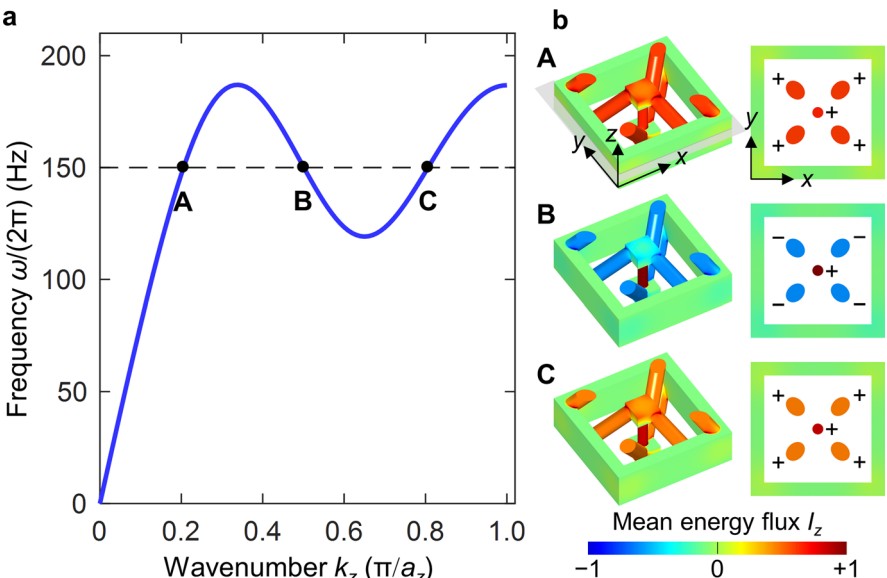

**Fig. 6 Roton-like behavior for airborne sound.** We consider a structure which is the complement of the one shown in Fig. 3. This leads to a network of channels inside a rigid material in which air can flow. **a** Acoustical dispersion relation $\omega(k_z) = \omega(-k_z)$ for air pressure waves propagating along the $z$-direction, exhibiting a roton-like minimum within the first Brillouin zone $|k_z| \leq \pi/a_z$. Higher bands at (much) higher frequencies are not shown. **b** Mean energy flux $I_z$ on a false-color scale along the $z$-direction for the three eigenmodes **A**, **B**, and **C** marked in **a** at the same frequency of 150 Hz. As in Fig. 4b, backward-wave behavior is observed for mode **B**, for which the group velocity is negative. The lattice constant is chosen as $a_z = 10$ cm. All other parameter ratios are the same as in Fig. 3. We assume ambient conditions, corresponding to an airborne speed of sound of $v_{air} = 340$ m/s and a mass density of $\rho_{air} = 1.29$ kgm$^{-3}$.

The approach can be generalized to more than just two types of interactions, i.e., to the combined action of nearest-neighbor, second-nearest-neighbor, third-nearest-neighbor, etc. interactions. This generalization would allow for tailoring almost any wanted dispersion relation of acoustical waves or phonons in the spirit of a Fourier expansion. However, feasible corresponding three-dimensional microstructures would need to be designed. The idea of beyond-nearest-neighbor interactions can also be combined with a variety of established other mechanisms in metamaterial design to obtain further unusual and useful effective material behaviors.

## Methods

**Energy flow in the 1D toy model**. Equation (3) for the energy flow in the 1D toy model has been derived as follows. For the Bloch waves with wavenumber $k$ and angular frequency $\omega$, the displacement of the mass at lattice site $n$ is given by $u_n(t) = \tilde{u} \exp(\mathrm{i}(kna - \omega t))$. The energy, which is transmitted through the Hooke's springs that couple neighboring masses (cf. Fig. 2a), averaged over an oscillation period, is given by

$$\langle p_1(k)\rangle = \frac{1}{2}\mathrm{Re}\left\{K_1\left(u_{n-1} - u_n\right)\frac{\mathrm{d}u_n^*}{\mathrm{d}t}\right\} = \frac{1}{2}\tilde{u}^2\omega K_1\sin(ka). \quad (4)$$

Herein, the term $K_1\left(u_{n-1} - u_n\right)$ is the force acting onto the mass at lattice site $n$ by the Hooke's spring to its left and the symbol $*$ stands for the complex conjugate. Similarly, the mean energy flow through the Hooke's springs that mediate the beyond-nearest-neighbor coupling is

$$\langle p_N(k)\rangle = \frac{1}{2}\mathrm{Re}\left\{K_N\left(u_{n-N} - u_n\right)\frac{\mathrm{d}u_n^*}{\mathrm{d}t}\right\} = \frac{1}{2}\tilde{u}^2\omega K_N\sin(Nka). \quad (5)$$

The total mean energy flow through the 1D toy model is the sum of these two contributions

$$\langle p(k)\rangle = \langle p_1(k)\rangle + N\langle p_N(k)\rangle = \frac{1}{2}\tilde{u}^2\omega(K_1\sin(ka) + NK_N\sin(Nka)). \quad (6)$$

**Elastic metamaterials**. We have numerically solved the eigenvalue equation derived from linear Cauchy elasticity[23] for the displacement vector field $\vec{u}_{\vec{k},i}(\vec{r})$ with band index $i$ at wave vector $\vec{k}$ and for the angular frequency $\omega_i(\vec{k})$

$$\frac{E}{2(1+\nu)(1-2\nu)}\vec{\nabla}\left(\vec{\nabla}\cdot\vec{u}_{\vec{k},i}(\vec{r})\right) + \frac{E}{2(1+\nu)}\vec{\nabla}^2\vec{u}_{\vec{k},i}(\vec{r}) = -\rho\omega_i^2(\vec{k})\vec{u}_{\vec{k},i}(\vec{r}) \quad (7)$$

by using the commercial software Comsol Multiphysics, its MUMPS solver, Floquet-Bloch periodic boundary conditions corresponding to the three-dimensional geometry shown in Fig. 3 for all three spatial directions, and traction-free boundary conditions for all interfaces to voids (air or vacuum). $E$ is the Young's modulus, $\nu$ the Poisson's ratio, and $\rho$ the mass density of the constituent material. The geometry shown in Fig. 3 has been meshed by about 100 thousand tetrahedra to achieve convergence of the results. The energy flux vector averaged over one temporal oscillation period has been evaluated by the formula

$$\vec{I}_i(\vec{k}) = \frac{1}{2}\mathrm{Re}\left\{-\frac{E\nu}{(1+\nu)(1-2\nu)}\left(\vec{\nabla}\cdot\vec{u}_{\vec{k},i}(\vec{r})\right)\frac{\mathrm{d}\vec{u}_{\vec{k},i}^*(\vec{r})}{\mathrm{d}t}\right.$$
$$\left.-\frac{E}{2(1+\nu)}\left(\vec{\nabla}\vec{u}_{\vec{k},i}(\vec{r})\cdot\frac{\mathrm{d}\vec{u}_{\vec{k},i}^*(\vec{r})}{\mathrm{d}t} + \vec{u}_{\vec{k},i}(\vec{r})\vec{\nabla}\cdot\frac{\mathrm{d}\vec{u}_{\vec{k},i}^*(\vec{r})}{\mathrm{d}t}\right)\right\}, \quad (8)$$

where $*$ denotes the complex conjugate. In Fig. 4b, the $z$-component of this vector, $I_z$, is plotted. This component has been obtained directly from the Solid Mechanics Module of Comsol Multiphysics.

**Metamaterials for airborne acoustical waves**. In the calculations shown in Fig. 6b, we have solved the scalar wave equation[22] for the air pressure modulation $\widetilde{P}_{\vec{k},i}(\vec{r})$ of the band with band index $i$ at wave vector $\vec{k}$ in the Fourier domain, corresponding to the eigenvalue problem

$$\vec{\nabla}\cdot\left(\vec{\nabla}\widetilde{P}_{\vec{k},i}(\vec{r})\right) = -\frac{\omega_i^2(\vec{k})}{v_{\mathrm{air}}^2}\widetilde{P}_{\vec{k},i}(\vec{r}) \quad (9)$$

with the speed of sound in air $v_{\mathrm{air}} = 340$ m/s, by using the commercial software Comsol Multiphysics. The considered geometry is the complement of the geometry illustrated and defined in Fig. 3. We assume Bloch periodic boundary conditions along all three spatial directions and the walls of all channels as rigid immovable boundaries via Neumann boundary conditions. The energy flux vector averaged

over one temporal oscillation period has been evaluated by

$$\vec{I}_i(\vec{k}) = \frac{1}{2}\mathrm{Re}\left\{\frac{\mathrm{i}}{\omega_i(\vec{k})\rho_{\mathrm{air}}}\widetilde{P}_{\vec{k},i}(\vec{r})\vec{\nabla}\widetilde{P}_{\vec{k},i}^*(\vec{r})\right\}, \quad (10)$$

with the imaginary unit $\mathrm{i}$ and the air mass density $\rho_{\mathrm{air}} = 1.29$ kgm$^{-3}$. In Fig. 6b, the $z$-component of this vector, $I_z$, is plotted. For the numerical calculations, the Pressure Acoustics Module of Comsol Multiphysics has been used. The quantity $I_z$ has been obtained directly by this module.

## Data availability

The data that support the plots within this paper and other findings of this study are available from the corresponding author upon reasonable request.

## Code availability

Numerical simulations in this work for the 1D toy model are all performed using the commercial software MATLAB. Numerical simulations in this work for the elastic and acoustic metamaterials are all performed using the commercial software COMSOL Multiphysics. All related codes can be built with the instructions provided in the main text and in the Methods section.

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

## Acknowledgements

We thank Tobias Frenzel (KIT) and Jörg Schmalian (KIT) for stimulating discussions. Y.C. acknowledges support by the Alexander von Humboldt Foundation and by the National Natural Science Foundation of China (contract 11802017). This research has additionally been funded by the Deutsche Forschungsgemeinschaft (DFG, German Research Foundation) under Germany's Excellence Strategy via the Excellence Cluster "3D Matter Made to Order" (EXC-2082/1-390761711), which has also been supported by the Carl-Zeiss Foundation through the "Carl-Zeiss-Foundation-Focus@HEiKA", by the State of Baden-Württemberg, and by the Karlsruhe Institute of Technology (KIT). We further acknowledge support by the Helmholtz program "Materials Systems Engineering" (MSE). M.K. is grateful for support by the EIPHI Graduate School (contract ANR-17-EURE-0002). We acknowledge support by the KIT-Publication Fund of the Karlsruhe Institute of Technology.

## Author contributions

M.W. and M.K. had the idea to incorporate beyond-nearest-neighbor interactions, Y.C. and M.K. designed the metamaterials. Y.C. performed the numerical simulations. M.W. supervised the project and drafted the paper. All authors contributed to the interpretation of the results and to the writing of the manuscript.

## Funding

## Competing interests

The authors declare no competing interests.
