## [Peer Review File · Nature Communications]

REVIEWER COMMENTS

Reviewer #1 (Remarks to the Author):

Yi et al present a totally new concept that they call roton metamaterials and which is mainly a novel explanation of the roton anomalous dispersion in Helium 3 and 4 and Bose-Einstein condensates.

I find this paper extremely good, novel and important in physics. This is probably one of the major metamaterial contributions in the last years. This study should deserve a particular attention and will certainly attract many people in both physics and metamaterial communities.

This study, very clearly, shows the interaction order importance in the dispersion relation between different masses (or atoms in condensed matter) and present a clear design for experimental proof of the anomalous roton dispersion in elasticity and acoustics. Beyond the concept, it brings an answer to the confusion made by Feynman in his report.

I am sure it will attract many additional studies on the negative group velocities and on negative refraction out of resonance. It should pave the way for new functionalities in phononic crystals and beyond. It will give a new way to design and shape the first band of dispersion relations in acoustics, elasticity and even electromagnetism.

For these reasons, I recommend its publication in the current form and I think it should be advertised by Nature as a great contribution.

I would recommend to the authors to give an additional gif or video of the Figure 3 with different orientation angles to make it easier to understand.

Reviewer #2 (Remarks to the Author):

In the manuscript, the authors designed elastic and acoustic metamaterials for roton-like dispersion relations. This has been made possible by mixing the ordinary dispersion relation with the one of three-fold degeneracy through beyond-nearest-neighbor interactions. Roton-like dispersion relations together with the mode shapes were calculated numerically. The results are interesting to the field. This reviewer has a few questions that need to be addressed.

1. Roton-like dispersion relations have been demonstrated in the context of micropolar continuum elasticity (Ref. 25). The authors mentioned "Notably, effects of periodicity, such as Bragg reflection, as well as effects of local low-frequency resonances play strictly no role in micropolar elasticity theory and can hence not explain these findings [25]." Why effects of periodicity, as well as effects of local low-frequency resonances must be needed to explain these findings? Is it the reason that the authors suggested the metamaterial with beyond-nearest-neighbor interactions? Effects of periodicity have been studied in the manuscript, however, effects of local low-frequency resonances have not been touched.

2. Figure S1 is interesting. The reviewer noticed that the incidence can generate three modes simultaneously. Are the amplitudes of the three modes controllable? How the authors harness the mode with negative phase velocity for applications, i.e. negative refraction?

3. The title of the manuscript is "... 3D metamaterials". However, the authors only studied

waves propagating to one direction. What happens if waves change propagation directions?

4. What happens at the two points on the dispersion relations with zero group velocity?

5. Dispersion relations in Fig. 4a have two modes at low frequencies. One is longitudinal mode; the other is transverse mode. Are there any other modes at low frequencies, i.e. torsional mode?

6. The experimental validation will be nice if possible? Or mention what is the main challenge for experimental test?

Reviewer #3 (Remarks to the Author):

The authors produce an analog to Rotons in two acoustic metamaterial systems exploiting next-nearest neighbors. Although Rotons have been found in acoustic/elastic micropolar systems, this is the first work to find them in next-nearest neighbor systems. The analysis is rigorous and the explanations and methods are easily followed. The authors provide a careful consideration of how these systems could be built and the resulting symmetries. The discussion on chiral versus achiral, and how this affects the presence of Rotons, is particularly valuable. Unfortunately, the authors do not fabricate and test any of the designs presented and this will likely affect the impact of the work.

Response to Reviewers

We thank the reviewers for their comments. In what follows, *we repeat the comments in red and italics*, we respond in green, we cite from the original manuscript in black, and highlight changes made to the manuscript in blue.

Reviewer #1

Yi et al present a totally new concept that they call roton metamaterials and which is mainly a novel explanation of the roton anomalous dispersion in Helium 3 and 4 and Bose-Einstein condensates.

I find this paper extremely good, novel and important in physics. This is probably one of the major metamaterial contributions in the last years. This study should deserve a particular attention and will certainly attract many people in both physics and metamaterial communities.

This study, very clearly, shows the interaction order importance in the dispersion relation between different masses (or atoms in condensed matter) and present a clear design for experimental proof of the anomalous roton dispersion in elasticity and acoustics. Beyond the concept, it brings an answer to the confusion made by Feynman in his report.

I am sure it will attract many additional studies on the negative group velocities and on negative refraction out of resonance. It should pave the way for new functionalities in phononic crystals and beyond. It will give a new way to design and shape the first band of dispersion relations in acoustics, elasticity and even electromagnetism.

For these reasons, I recommend its publication in the current form and I think it should be advertised by Nature as a great contribution.

We are very grateful to reviewer #1 for his/her overwhelmingly positive assessment of our manuscript. We also thank the reviewer for accepting our paper in its current form and for the recommendation to advertise it as a great contribution by *Nature*.

I would recommend to the authors to give an additional gif or video of the Figure 3 with different orientation angles to make it easier to understand.

In the revised version, we provide a video (referred to as Supplementary Movie 1) to yet better visualize the beyond-nearest-neighbor interactions in the bulk metamaterial (as shown in Fig. 3(b)) from different perspectives. The caption of Fig. 3 now reads:

“... The part highlighted in red illustrates the beyond-nearest-neighbor interaction. Two red rods connect a first cube to the red frame (made partly transparent at the corner). Two further red rods connect this frame to a second cube, which has a distance $3a_z$ with respect to the first cube. An animated view of the structure is given in Supplementary Movie 1.”

Reviewer #2

In the manuscript, the authors designed elastic and acoustic metamaterials for roton-like dispersion relations. This has been made possible by mixing the ordinary dispersion relation with the one of three-fold degeneracy through beyond-nearest-neighbor interactions. Roton-like dispersion relations together with the mode shapes were calculated numerically. The results are interesting to the field.

We thank reviewer #2 for his/her positive assessment of our work and for pointing out that the results are interesting for the field.

This reviewer has a few questions that need to be addressed.

1. Roton-like dispersion relations have been demonstrated in the context of micropolar continuum elasticity (Ref. 25). The authors mentioned “Notably, effects of periodicity, such as Bragg reflection, as well as effects of local low-frequency resonances play strictly no role in micropolar elasticity theory and can hence not explain these findings [25].” Why effects of periodicity, as well as effects of local low-frequency resonances must be needed to explain these findings? Is it the reason that the authors suggested the metamaterial with beyond-nearest-neighbor interactions? Effects of periodicity have been studied in the manuscript, however, effects of local low-frequency resonances have not been touched.

The sole purpose of this paragraph is to prepare the reader that the mechanisms leading to roton-like dispersion relations in reference [25] and those in our present work are distinct. Extraordinary Bragg reflections due to beyond-nearest-neighbor interactions are the mechanism in our work. Bragg reflections are not accounted for on the level of Eringen micropolar continuum elasticity in reference [25]. While the statement on local resonances in our original manuscript is technically correct, too, it is indeed not really relevant here and has obviously led to confusion. We have therefore taken it out in the revised version of our manuscript.

The revised part of the manuscript reads:

“... In sharp contrast, it has recently been shown that **chiral** Eringen micropolar continuum elasticity theory²⁴ can lead to roton-like dispersion relations for transverse acoustical elastic waves²⁵. **In their work²⁵, chirality has been a necessary mechanism, whereas mechanisms based on periodicity, such as ordinary or extraordinary Bragg reflections, are not accounted for in micropolar elasticity theory²⁴. More broadly speaking,** mechanisms such as ordinary Bragg reflection^{26, 27}, local resonances²⁸⁻³¹, near-ideal joints³²⁻³⁴ introducing soft modes, spatial or temporal symmetry breaking³⁵⁻³⁸, topology^{39, 40}, duality^{41, 42}, as well as geometrical nonlinearities^{34, 43} have independently given rise to a wealth of other unusual dispersion relations and quasi-static behaviors of elastic and acoustical metamaterials. ...”

With this revision, we have also prepared the reader that reference [25] found roton-like dispersion relations for chiral media only, whereas chirality is not necessary in our case of beyond-nearest-neighbor interactions.

In addition, the preprint [25] has meanwhile appeared in *Physical Review Letters*. We have updated this reference correspondingly.

2. Figure S1 is interesting. The reviewer noticed that the incidence can generate three modes simultaneously. Are the amplitudes of the three modes controllable? How the authors harness the mode with negative phase velocity for applications, i.e. negative refraction?

Yes, the amplitudes of the three modes are indeed controllable. Previously, we have not addressed this point. Following this point of reviewer #2, we have added the new Supplementary Fig. S3 to our manuscript. Herein, the considered setup is the same as in Supplementary Fig. S1, but we do not only

excite the mass at site $n = 0$ but also its two immediate neighbors to the left and right. We find that each of the three modes propagating to the right in Supplementary Fig. S1 can be induced *separately* by adequately choosing the excitation conditions.

This new figure and its caption read:

“

Figure S3 | Controlling the coupling to the right-moving wave packets. As Fig. S1, but we do not only excite the center mass at site $n = 0$ as in Fig. S1 but also its two neighbor masses to the left and right at sites $n = -1$ and $n = +1$, respectively. As a result, only a single wave packet (rather than a triplet as in Fig. S1) propagates to the right-hand side in each of the three cases (a)-(c). For case (a), only a single wave packet with negative phase velocity propagating to the right-hand side emerges. The three dashed black lines for $x > 0$ in (a)-(c) are guides to the eye and refer to the three wave packets in Fig. S1. In (a)-(c), we use the excitation conditions $u_0(t) = \tilde{u}_0 \cos(\omega t) \exp(-(t/\tau)^2)$, $u_{-1}(t) = \tilde{u}_{-1} \cos(\omega t + \phi_{-1}) \exp(-(t/\tau)^2)$, and $u_{+1}(t) = \tilde{u}_{+1} \cos(\omega t + \phi_{+1}) \exp(-(t/\tau)^2)$. As in Fig. S1, we choose $\omega = 0.5 \omega_0$ and $\tau = 100/\omega$. (a) $\tilde{u}_{-1} = \tilde{u}_{+1} = 0.5 \tilde{u}_0$, $\phi_{-1} = -\pi/2$, and $\phi_{+1} = +\pi/2$. (b) $\tilde{u}_{-1} = \tilde{u}_{+1} = \tilde{u}_0$, $\phi_{-1} = -0.8 \pi$, and $\phi_{+1} = +0.8 \pi$. (c) $\tilde{u}_{-1} = \tilde{u}_{+1} = \tilde{u}_0$, $\phi_{-1} = -0.2 \pi$, and $\phi_{+1} = +0.2 \pi$.”

In addition, we have also revised the main text connecting to Figs. S1 and S3:

“... For the latter, group and phase velocity have opposite sign (see insets in Supplementary Fig. S1). These findings are consistent with the expectation from Fig. 1(b) and confirm our reasoning. Furthermore, Supplementary Fig. S3 shows that each of the three right-propagating modes can be excited selectively by tailoring of the excitation conditions. ...”

The numbers of the other Supplementary figures have been changed accordingly.

As to the reviewer’s question on negative refraction, it is clear that refraction does not occur in a one-dimensional system discussed so far. Refraction does occur in three-dimensional systems, which brings us to the next question of the reviewer.

3. The title of the manuscript is “... 3D metamaterials”. However, the authors only studied waves propagating to one direction. What happens if waves change propagation directions?

Indeed, for the three-dimensional structure discussed in Fig. 3, we had only shown the results of wave propagation along the z -direction in Fig. 4 in our manuscript. This choice was motivated by the direct connection to the one-dimensional toy model.

Following the suggestion of the reviewer, we have now also included other directions in three dimensions in the more complete band structure in the new Supplementary Fig. S4 and its caption:

Figure S4 | Elastic metamaterial phonon band structure. As Fig. 4 (for the metamaterial structure shown in Fig. 3), but for many high-symmetry directions rather than only the Γ Z or z-direction as in Fig. 4. (a) Illustration of the first Brillouin zone of the tetragonal-symmetry real-space lattice and selected high-symmetric directions in reciprocal space (marked in blue). (b) Calculated three-dimensional phonon band structure with the characteristic directions as indicated in (a). Clearly, due to the used tetragonal symmetry, roton-like acoustical dispersion relations only occur for the Γ Z direction. The corresponding colored bands (blue and red) are the same as the ones shown in Fig. 4.”

We refer to this new figure S4 in the main part of the paper in the modified caption of Fig. 4:

“... They partly result from local resonances within the unit cell, leading to finite values of ω at zero wavenumber $k_z = 0$. The complete phonon band structure for all high-symmetry directions in three dimensions is shown in Supplementary Fig. S4. (b) Mean energy flux I_z ...”

Furthermore, we should like to emphasize that we have clearly highlighted the crucial importance of the three-dimensionality of the metamaterial in the original version of the main text:

“ ... **Three-dimensional microstructured elastic metamaterial**

Next, we translate the behavior of the 1D mathematical toy model into a practical metamaterial structure. From Fig. 2(a) it is clear that the (red) beyond-nearest-neighbor springs unavoidably overlap in two dimensions, making it necessary to go to three dimensions. The 3D architecture depicted in Fig. 3 is ...”

Finally, it is conceivable that 3D metamaterials with cubic symmetry may also exhibit roton-like acoustical dispersion relations. However, the design of such more symmetric metamaterials or of metamaterials with completely isotropic properties is way beyond the scope of the present paper.

4. What happens at the two points on the dispersion relations with zero group velocity?

We are not quite sure we understand the question. At these points, the group and the energy velocity become zero because the positive energy transported through the nearest-neighbor springs and the negative energy transported through the third-nearest-neighbor springs add up to zero. We believe that this vortex-like energy flow has been explained in sufficient detail in the section on the 1D toy model. Zero group and energy velocity are just a special case of the more general discussion. To improve our paper for the reader in this regard, this aspect is now explicitly mentioned in the revised version of the manuscript:

“... This leads to a positive phase velocity in the interval $k \in [0, \pi/a]$, whereas the group velocity and the mean energy flow are negative for part of this interval (approximately for $k \in [\pi/(3a), 2\pi/(3a)]$) for $K_N/K_1 > 1/N$. This behavior corresponds to a backward wave. The two wavenumbers for which the total energy flow and hence the group velocity are zero (cf. Fig. 2(b)) are merely special cases. On the basis of this discussion, ...”

Furthermore, as usual, zero group velocity is connected to peaks in the phonon density of states. This aspect has been pointed out in the introduction of our manuscript:

“... Second, the extrema of the dispersion relation in Fig. 1 correspond to zero group velocity and hence to peaks in the wave density of states. ...”

5. Dispersion relations in Fig. 4a have two modes at low frequencies. One is longitudinal mode; the other is transverse mode. Are there any other modes at low frequencies, i.e. torsional mode?

As usual, torsional or twist modes do not occur in bulk crystals, that is, in structures that are periodic and infinite along all three spatial directions. Mathematically, torsional or twist modes are suppressed by the used Floquet-Bloch boundary conditions along all three spatial directions for the bulk case.

However, such additional torsional or twist modes do occur in beam structures. A beam structure has indeed been considered in our Supplementary Fig. S3 (now Supplementary Fig. S5). The black twist band is explicitly mentioned in the caption. Perhaps, the reviewer has overlooked this band. To emphasize this aspect for the reader in the main text of our paper, we have modified it correspondingly:

“... Therefore, in Supplementary Fig. S5, we show results for a beam with a cross section of merely 2×2 unit cells (cf. Fig. 3(a)). A roton-like dispersion relation for the transverse bands is maintained. Roton-like behavior is also found for the twist band in Supplementary Fig. S5, which additionally appears due to the finite cross section of the beam.

Furthermore, we have emphasized ...”

6. The experimental validation will be nice if possible? Or mention what is the main challenge for experimental test?

This point is closely connected to the last sentence of reviewer #3 (see below). In regard to both, the editor has communicated to us that an experimental validation is not strictly necessary for publication in *Nature Communications*.

In fact, one should be aware that we are not just suggesting a single metamaterial blueprint in the present paper but rather four different blueprints: A bulk achiral elastic metamaterial, a bulk chiral elastic metamaterial, a bulk metamaterial for airborne sound, and an elastic metamaterial beam with finite cross section. Presenting experimental validations for all of these would completely go beyond the scope (and length) of a *Nature Communications* paper. Presenting only an experimental validation for one of the blueprints would appear very strange to us. In any case, for none of these four cases experimental validations are presently available.

As to this question of reviewer #2, the main challenge does not lie in the manufacturing of these metamaterial samples; it lies in the band-structure measurements. We discuss this aspect more clearly in the revised version of the conclusions of our present paper:

“... The third-nearest-neighbor interaction gives rise to a hybridization of phonon branches with different spatial periods and hence to extraordinary Bragg reflections with reciprocal lattice vectors smaller than the wave vector at the edge of the first Brillouin zone. For both, the proposed (achiral and chiral) 3D microscopic microstructures for elastic waves and the proposed 3D macroscopic channel-

based structures for airborne sound waves, the 3D additive manufacturing technology required to make the metamaterial unit cells is readily available. However, large numbers of unit cells are needed to avoid edge effects. This aspect together with directly measuring the roton-like dispersion relations represents a challenge.

The approach can be generalized to more than just two types of interactions, i.e., ...”

Reviewer #3

The authors produce an analog to Roton in two acoustic metamaterial systems exploiting next-nearest neighbors. Although Roton have been found in acoustic/elastic micropolar systems, this is the first work to find them in next-nearest neighbor systems. The analysis is rigorous and the explanations and methods are easily followed.

The authors provide a careful consideration of how these systems could be built and the resulting symmetries. The discussion on chiral versus achiral, and how this affects the presence of Roton, is particularly valuable.

We thank the reviewer for his/her positive comments on our paper and his/her appreciation of the discussion on chiral structures versus achiral structures. Indeed, our design solely relies on beyond-nearest-neighbor interactions and does not necessarily require chirality, which is a necessary mechanism for obtaining roton-like dispersion relations in micropolar continuum elasticity (cf. Ref. [25]). This aspect has also been discussed in detail in our response to point 1. of reviewer #2 (see above).

Unfortunately, the authors do not fabricate and test any of the designs presented and this will likely affect the impact of the work.

This aspect is closely related to point 6. of reviewer #2 and we kindly refer reviewer #3 to our above response and to the corresponding changes made to the conclusions of the paper.

We feel that our systematic discussion of the effects of beyond-nearest-neighbor interactions in four different metamaterial systems will stimulate further theoretical studies and experiments in this direction and thereby lead to a sufficiently large impact.

The overwhelmingly positive response of reviewer #1 makes us optimistic that this assessment will prove true: *“... probably one of the major metamaterial contributions in the last years ...”, “... it will attract many additional studies on the negative group velocities and on negative refraction out of resonance. It should pave the way for new functionalities in phononic crystals and beyond. It will give a new way to design and shape the first band of dispersion relations in acoustics, elasticity and even electromagnetism.”*

REVIEWER COMMENTS

Reviewer #2 (Remarks to the Author):

The revised addressed most of my questions properly. However, I suggest to include Fig. 4S in the main manuscript to make readers have a better picture on this design.

Response to the Reviewer

In what follows, *we repeat the comments of the single reviewer (reviewer #2) in red and italics*, we describe the changes made to the manuscript and the Supplementary Information in green, we cite from the original manuscript in black, and highlight changes made to the manuscript in blue.

Reviewer #2

The revised addressed most of my questions properly. However, I suggest to include Fig. 4S in the main manuscript to make readers have a better picture on this design.

We thank the reviewer for this positive response. As suggested by the reviewer, we have shifted the 3D band structure shown in Supplementary Figure 4 (originally referred to as Fig. S4) from the Supplementary Information into the main paper, where it is now Fig. 5. The old Fig. 5 is now Fig. 6. We have renumbered Supplementary Figure 5 to 7 accordingly, too.

The new Fig. 5 needs to be addressed within the main text (previously, we addressed Supplementary Figure 4 only in the caption of Fig. 4). We have removed the sentence “The complete phonon band structure for all high-symmetry directions in three dimensions is shown in Supplementary Fig. S4.” from the caption of Fig. 4. On page 8 of the main text, we have inserted the following two new sentences:

“... In his work on rotons⁴, Feynman referred to the backward energy flow contribution as a “return flow”.

The complete phonon band structure for all high-symmetry directions in three dimensions is shown in Fig. 5. Roton behavior only occurs along the Γ Z-direction (cf. Fig. 4(a)).

The roton-like behavior discussed thus far refers to the bulk ...”